# Prevalence of hepatitis B virus infection and associated risk factors among pregnant women attending antenatal clinic in Mulago Hospital, Uganda: a cross-sectional study

Simon Peter Kayondo [1,2] Josaphat K Byamugisha,[3] Peter Ntuyo[3]

[1]Obstetrics and Gynaecology, Makerere University College of Health Sciences, Kampala, Uganda
[2]Association of Obstetricians and Gynaecologists of Uganda, Uganda
[3]Obstetrics and Gynecology, Makerere University/Mulago National Referral Hospital, Kampala, Uganda

**Correspondence to**
Dr Simon Peter Kayondo;
kayondosimondr@gmail.com

## ABSTRACT

**Aim** To determine the prevalence and factors associated with hepatitis B virus infection among pregnant women attending antenatal clinic in Mulago Hospital.
**Design** Cross-sectional observational study.
**Setting** Mulago National Referral Hospital, Uganda, antenatal clinic.
**Participants** We randomly selected 340 pregnant women attending their first antenatal visit at Mulago Hospital antenatal clinic.
**Primary outcome** Hepatitis B surface antigen positivity.
**Results** We recruited 340 participants, with a mean age of 27±5.7 years, and a median gravidity of 3. The prevalence of hepatitis B virus infection among pregnant women attending the antenatal clinic in Mulago Hospital, in our study, was 2.9% (95% CI 1.58% to 5.40%, n=10). Factors positively associated with hepatitis B virus infection were: marital status (adjusted OR (aOR)=11.37, p=0.002), having a hepatitis B positive family member (aOR=49.52, p<0.001) and having had a blood or body fluid splash to mucous membranes from a hepatitis B positive patient (aOR=61.69, p=0.015). Other factors such as age, socioeconomic status, number of sexual partners, HIV serostatus, piercing of ears and history of blood transfusion were not significantly associated with hepatitis B virus infection in this study.
**Conclusion** The prevalence of hepatitis B virus infection among pregnant women attending antenatal clinic in Mulago Hospital was of intermediate endemicity. We found that marital status, having a hepatitis B positive family member at home and having had a blood or body fluid splash to mucous membranes from a hepatitis B positive patient were independently associated with hepatitis B infection. Factors such as age, HIV status, history of blood transfusion, piercing of ears and social status were not associated with hepatitis B status in this study.

## INTRODUCTION

Hepatitis B is a viral infection of the liver that can cause acute and chronic disease. Worldwide, about 2 billion people have been exposed to hepatitis B, and about 257 million people are living with hepatitis B infection (defined as hepatitis B surface antigen (HBsAg) positive).[1] Hepatitis B infection resulted in 887 000 deaths

---

**Strengths and limitations of this study**

► We followed standard laboratory procedures at an ISO 9001 certified laboratory using a highly sensitive and specific hepatitis B surface antigen test kit performed by qualified personnel.
► Laboratory technologists performed all testing blinded through use of coded specimens.
► We recruited the study participants using a random sampling procedure.
► Our study setting was a national referral and teaching hospital, hence our findings may not be generally applicable in other settings.
► Because of limited number of positive cases in our study, we had limited power to assess the significance of associated factors.

---

in 2015, mostly from liver complications: liver cirrhosis and hepatocellular carcinoma. Globally, hepatitis B prevalence in adults is highest in the WHO Western Pacific Region (6.2%) and the WHO African region (6.1%). In Sub-Saharan Africa, the overall HBsAg carrier rate in the general population is 5%–20%, which is among the highest in the world. Perinatal transmission is estimated at 1%–5%.[2] Seventy-two per cent of the Ugandan population are exposed to hepatitis B in their lifetime, leading to a national prevalence of hepatitis B infection of approximately 10%. There is regional variation within Uganda, however, the highest prevalence is in the northern part of the country, ranging from 19% in the northwest to 25% in the northeast.[3 4] It is not known how many pregnant women are hepatitis B positive worldwide. However, in Northern Uganda, prevalence is around 11.8%.[5] Allen *et al* estimated the prevalence among the antenatal population of Mulago Hospital at 0.9% in 2016,[6] although this study had several limitations with regard to

participant recruitment and estimation of mother-to-child transmission risk.

In highly endemic areas, vertical transmission of hepatitis B from mother to child at birth plays a key role. However, horizontal transmission in childhood also occurs.[1 7] Infants infected from their mothers, or before the age of 5 years, often develop chronic infection.[1 8 9] Hepatitis B can also be transmitted through percutaneous or mucosal exposure to infected blood or body fluids such as saliva, menstrual, vaginal and seminal fluids (including in men who have sex with men).[1 10] Reuse of needles and intravenous drug use are also high-risk activities for hepatitis B transmission. The infection may be transmitted through unsafe medical, dental and surgical procedures, and through practices such as tattooing or use of sharp instruments contaminated with blood.[1]

The risk of developing chronic hepatitis B for babies born to a mother with hepatitis B is greater than 90%, if not identified and treated at birth. It is therefore imperative for pregnant women to know their hepatitis B status, ideally through testing in the first trimester.[1] If the mother is hepatitis B positive, the baby should receive both the first dose of the hepatitis B vaccine and hepatitis B immunoglobulin at birth. This treatment is recommended by the Centres for Disease Control and Prevention (CDC), although it may not be available in all countries.[1 11 12] If treatment is given promptly within the first 12 hours of life, the newborn has >90% chance of lifelong protection against hepatitis B infection.[11–13] Infant vaccination for hepatitis B was introduced in Uganda in 2002,[4] as part of the Uganda National Expanded Programme for Immunisation schedule. However, adults born prior to 2002 (currently aged >18 years) remain a high-risk group.

Acute or chronic hepatitis B infection in pregnancy is generally similar to that in the general population. Hepatitis B in pregnancy is not teratogenic and does not increase maternal or fetal mortality.[14 15] However acute infection is associated with a higher incidence of low birth weight and prematurity than in the general population.[15] In pregnancy, the immune response against hepatitis B is less effective, possibly due to a shift from the Th1 to Th2 response. The WHO defines hepatitis B infection in newborns as HBsAg positivity 6 months after birth.[1 16] The risk of mother-to-child transmission is very high in the absence of prophylaxis, varying with the hepatitis B envelope antigen (HBeAg)/anti-HBe status of the mother: 70%–90% for HBeAg positive, 25% for HBeAg-negative/hepatitis B envelope antibody (HBeAb)-negative mothers and 12% for HBeAg-negative/anti-HBe-positive mothers.[13 15 17]

We aimed to determine the prevalence and risk factors associated with hepatitis B infection among pregnant women attending for antenatal care at Mulago Hospital in Kampala, Uganda.

## METHODS

### Study design and setting

This was a cross-sectional study, conducted at Mulago Hospital antenatal clinic. Mulago Hospital was founded in 1917 in Kampala City, the capital of Uganda, and is the Ugandan national referral hospital. It is a public general and teaching hospital, with a bed capacity of over 1500 beds. Mulago Hospital is located on Mulago hill, approximately 5 km northeast of Kampala's central business district. Approximately 2500 women attend for antenatal care at Mulago Hospital every month.

We purposively selected Mulago Hospital because it serves people from all ethnic groups and regions of Uganda. It also serves a high number of refugees, such as those from Rwanda, South Sudan and Congo.

Kawempe General Hospital, in Kawempe division, temporarily houses the Mulago Hospital antenatal clinic, as the main Mulago Hospital is undergoing major renovation. The clinic continues to offer preventive and curative services, such as health education, immunisation, prenatal maternal and fetal monitoring. Between 200 and 250 pregnant women visit the clinic on every Tuesday, Wednesday and Thursday, of whom about 100 each day are attending their first antenatal visit. We enrolled only mothers attending their first visit into the study, as the flow of first visit attendees in the antenatal care (ANC) clinic was convenient for recruitment, unlike their repeat visit counterparts. We enrolled women after their health education session, as they awaited examination.

### Study population

The study population was pregnant women attending antenatal clinic, between December 2018 and February 2019, in Mulago Hospital, Kawempe site.

### Eligibility criteria

#### Inclusion criteria

All pregnant women who were attending their first antenatal visit at Mulago Hospital, who were willing to participate in the study. Women below 18 years were eligible for recruitment as emancipated minors.

#### Exclusion criteria

We excluded women who could not understand either English or Luganda, which were the two languages used for the consenting process and administration of the questionnaires.

### Research variables

The dependent variable was hepatitis B infection among pregnant women, assessed by HBsAg positivity.

The independent variables were factors associated with positive hepatitis B status. These included the sociodemographic factors such as age, marital status, level of education, socioeconomic status and sexual behaviour. We also included obstetric factors such as gravidity and parity and other factors such as history of blood transfusion, history of caring for a hepatitis B positive patient, immunisation status, ear piercings, tattoos and intravenous drug use in our model.

## Sample size estimation

We estimated the sample size for prevalence using the Wiegand[18] Leslie formula, 1965.[19]

Using a prevalence from Northern Uganda of 11.8%,[5] the required sample size to determine prevalence would be 160 women.

We based the sample size estimates for factors associated with hepatitis B infection in pregnancy on the hypothesis that the prevalence of infection would be higher among women ≤20 years than among those >20 years, as found by Bayo *et al*.[5] We compared the proportions in subpopulations (Fleiss,[20] Statistical Methods for Rates and Proportions, formulas pages 3.18 and 3.19) using an OpenEpi calculator (accessed at http://www.openepi.com/SampleSize/SSCC.htm). Assuming power of 80%, type 1 error of 5% and frequency of hepatitis B infection of 20% among women ≤20 years and 8.7% among women >20 years,[5] we calculated a total sample size required for risk factor modelling of 340. Three hundred and forty women were included in the study.

## Sampling procedure

We used a random probability sampling method. During the day of data collection, we used the register book for the antenatal clinic attendance and the booking cards for the first visit attendees to identify eligible potential participants. We invited every 10th eligible woman to take part in the study. If the 10th woman declined participation in the study, then the next eligible woman was invited. There was an average daily attendance of ~100 eligible potential participants on 3 days per week. Data were collected over a 3-month period (December 2018 and February 2019).

## Study procedure

The antenatal clinic midwives, who were the research assistants, approached potential participants and explained the study purpose to them. They then screened them for eligibility and obtained informed consent for participation in the study. We used interviewer-administered semi-structured questionnaires to get sociodemographic information including age, gestational age, gravidity, parity, education and occupation, and other factors that could influence infection with hepatitis B virus such as injection drug use, tattoos, family history of hepatitis B infection, sexual history, among others. Obstetric factors like gravidity and parity, previous history of taking care of a hepatitis B positive patient, immunisation against hepatitis B were all assessed.

The laboratory technician collected 4 mL of whole blood from the antecubital fossa under aseptic technique, into BD Clot Activator Tube vacutainers (labelled with the participant's study number), using WHO (2010) guidelines on blood collection.[21] The technician then took the blood to the Kawempe Hospital laboratory, centrifuged and transferred an aliquot of serum, about 2 mL, into cryovials. We performed immunochromatographic assay using HBsAg rapid test strip, FaStep, for HBsAg testing following the manufacturer's instructions. This rapid test kit has a relative sensitivity of >99.9%, a relative specificity of 99.9% and an accuracy of >99.9%. We did quality control on this test kit using known positive and negative samples from MBN Clinical Laboratories, an ISO 9001 certified laboratory, before it was used to test any participant's samples. We performed the HBsAg rapid tests using the following procedure: the laboratory technician labelled a rapid test strip with the patient identifiers, and placed it on a flat surface. He pipetted three drops of serum from the cryovial and applied it to the sample pad of the test strip. We read the result after 15 min. A positive result was one where two-coloured bands appeared on the membrane, one in the control region and another in the test region. A negative result was one where only one band appeared in the control region, and none in the test region. An invalid result was one where a band failed to appear in the control region.

We produced results and gave a copy to the patient after being counselled by the principal investigator and the nurse counsellor. We sent all HBsAg positive samples to MBN Clinical Laboratories, Kampala, an ISO 9001 accredited laboratory, for confirmatory HBsAg testing, HBeAg testing and liver function tests. We took 5% of all HBsAg negative samples to MBN, for external quality assurance. We packed and transported all samples at 2–8°C, on the same day of collection. We referred hepatitis B positive mothers to the hepatitis B clinic in Mulago Hospital (Kiruddu) for further care, where the physicians performed their viral load assays and further assessments for treatment. We planned to vaccinate their babies against hepatitis B within 24 hours of birth. The study endeavoured to meet the cost of the HepB-BD vaccine for all study participants who were hepatitis B positive.

## Data management and analysis

The principal investigator double-entered the raw study data into Epidata V.3.1 and imported into STATA V.14.1. Numerical results are presented as frequencies or percentages. To assess factors associated with hepatitis B infection in pregnancy, we performed bivariate analysis and computed crude ORs and 95% CIs. Student's t-test was used to assess statistical significance between groups for continuous variables, while the Fischer's exact test was used for categorical variables.

To assess the independent association of these risk factors, we performed multivariate analysis. All independent variables with a p value of <0.2 at bivariate analysis and those with biological significance were included in multivariate logistic regression. Stepwise regression method was used to determine the model of best fit. We considered all associations with p value <0.05 as statistically significant.

## Ethical considerations

We explained information regarding the study to the subjects and got written consent for participation in the study in English/Luganda. We interviewed participants and collected samples, following local guidelines

regarding research participant privacy. Only study numbers, with unique participant identifiers, were used to label the questionnaires and the samples. We kept filled questionnaires separate from signed consent forms and locked away securely all study material, with only access to study team. Participants were free to drop out at any point of the study, with an assurance that if they did so this would not affect their ongoing antenatal care.

All researchers in the study were fully immunised against hepatitis B. They wore protective gear such as gloves, masks and eye glasses during laboratory procedures. These measures protected both mothers and health workers from acquiring hepatitis B infection from each other during the study.

### Patient and public involvement
While there were patients involved in this study as study participants, we did not involve them at the stage of the study design. However, the intended outcome in favour of the patient was that we made available hepatitis B birth dose vaccination for all exposed babies. We circulated a summary of the findings using SMS messaging to all study participants.

### RESULTS
### Description of the study population
We recruited 340 participants into the study, with a mean age of 27±5.7 years (table 1). 56.8% were from Kampala, 40% from Wakiso and 3.2% from elsewhere. Most of the participants (76.7%) had a secondary education and above, and 52.4% were employed. A little over a half of the participants (51.7%) were business people, 41.5% in professional jobs, 5.1% casual labourers, while 3% were health workers. More than half (65.3%) were of low socioeconomic status, earning less than 500 000 Uganda shillings per month. The majority of the participants (70.3%) had their sexual debut during their teenage years.

Over half of the participants were employed, the majority (89.4%) were living with a partner and most of them had more than one lifetime sexual partner (table 1).

### Factors associated with hepatitis B virus infection
Only 2.9% of study participants were HIV positive (table 2). Vaccination against hepatitis B was low, with 2.6% and 3.5% as the childhood and adult vaccination rates, respectively (table 2). About 2.1% of the participants reported having a hepatitis B positive family member at home (table 2).

### Prevalence
Ten out of 340 participants were HBsAg positive, comprising 2.9% (95% CI 1.58% to 5.40%).

All the HBsAg positive participants were HBeAg negative and had normal liver function tests.

**Table 1** Demographic/gynaecological/obstetric/sexual characteristics of the study participants

| | Frequency (n=340 (%)) |
|---|---|
| **Demographic characteristics** | |
| Age (years); mean (SD) | 27 (5.7) |
| Age (years) categorised | |
| ≤20 | 28 (8.2) |
| >20 | 312 (91.8) |
| Employment status | |
| Yes | 178 (52.4) |
| No | 162 (47.6) |
| Marital status | |
| Living with a partner | 304 (89.4) |
| No stable partner | 36 (10.6) |
| **Gynaecological/obstetric/sexual characteristics** | |
| Gravidity; median (IQR) | 3 (2–3) |
| Gravidity categorised | |
| 1 | 67 (19.7) |
| 2–4 | 211 (62.1) |
| >4 | 62 (18.2) |
| Weeks of amenorrhea categorised | |
| >20 | 211 (62.1) |
| ≤20 | 54 (15.9) |
| Unknown | 75 (22.1) |
| Mother hepatitis B positive? | |
| Yes | 4 (1.2) |
| No | 126 (37.1) |
| I don't know | 210 (61.7) |
| Number of sex partners; median (IQR) | 2 (1–3) |
| Number of sex partners categorised | |
| 1 | 117 (34.6) |
| ≥2 | 221 (65.4) |

Mean (SD) for numerical data, n (%) for categorical data.

### Associations between the independent variables and hepatitis B status
In bivariate analysis, we found the participant's marital status, weeks of amenorrhea, mother's hepatitis B status significantly associated with hepatitis B status (table 3). These were included as independent variables in multivariable analysis together with age of the woman.

Having had a blood or body fluid splash to the mucous membranes from a hepatitis B positive patient and having a hepatitis B positive family member were significantly associated with hepatitis B status, at bivariate analysis (table 4). Both were included in multivariable analysis.

Marital status, having a hepatitis B positive family member and history of a blood or body splash to mucous membranes from a hepatitis B positive patient were all

**Table 2** Immunosuppression characteristics, vaccination history, familial exposure, occupational exposure risk of study participants and exposure to hepatitis B through sharp instruments, medical and surgical procedures

| | Frequency (n=340 (%)) |
|---|---|
| **Immunosuppression characteristics** | |
| HIV serostatus | |
| Positive | 10 (2.9) |
| History of diabetes mellitus | |
| Yes | 3 (0.9) |
| No | 47 (13.8) |
| Unknown | 290 (85.3) |
| **Vaccination history** | |
| Hepatitis B vaccination in the first year of life | |
| Yes | 9 (2.6) |
| No | 240 (70.6) |
| Don't know | 91 (26.8) |
| Hepatitis B vaccination after the age of 10 years | |
| Yes | 12 (3.5) |
| No | 299 (88.0) |
| Don't know | 29 (8.5) |
| **Exposure through sharp instruments/medical/surgical procedures** | |
| Ever had a blood transfusion | |
| Yes | 18 (5.3) |
| Ever pierced your ears | |
| Yes | 275 (80.9) |
| **Familial exposure** | |
| Have a hepatitis B positive family member at home? | |
| Yes | 7 (2.1) |
| No | 291 (85.6) |
| Don't know | 42 (12.3) |
| **Occupational exposure** | |
| Ever had needle stick injury? | |
| Yes | 4 (1.2) |
| Blood or body fluid splash from hepatitis B positive patient | |
| Yes | 2 (0.6) |

significantly independently associated with positive testing for hepatitis B (table 5). The association with age was not statistically significant (table 5).

## DISCUSSION

We conducted this study to determine the prevalence and risk factors associated with hepatitis B infection among pregnant women attending the antenatal clinic in Mulago Hospital, a multicultural, multiracial national referral hospital in urban Uganda.

Although Allen *et al*, in early 2016, estimated the prevalence of hepatitis B among pregnant women at 0.9%, before Mulago Hospital was transferred to the new Kawempe site,[6] their study was not without limitations. Notably, the prevalence of hepatitis B in their cohort was much lower, compared with our cohort (0.9% vs 2.9%). Apart from having a different sociodemographic profile, the 2016 study excluded pregnant women below 14 years and above 43 years. It also excluded all participants who were bedridden, and all those with obstetric emergencies. Application of these exclusion criteria may have skewed the participating population compared with the general obstetric population. A further explanation for the observed difference may be because Kawempe, the site at which we performed our study, is an area of significant socioeconomic deprivation, where high-risk activities such as prostitution are common. By contrast the 2016 study was performed at the old Mulago site, where the risk profile of the population is likely to be less. Nonetheless, since majority (60%) of all hepatitis B positive pregnant women present in the first half of the pregnancy, as shown by our study (table 3), it is an opportunity for early testing and intervention to reduce mother-to-child transmission.

The prevalence of hepatitis B in our study (2.9%) falls with the WHO 'intermediate endemicity' category (2%–8%).[22] It is comparable to the prevalence previously reported by Pirillo *et al* in Kampala (4.9%). Internationally, the prevalence in our study is also similar to two Rwandan studies that reported 2.4% and 3.7%, respectively.[23 24] However, the Rwandan studies were both conducted in HIV positive pregnant women only. Other studies from Kenya and Ethiopia respectively reported prevalences of hepatitis B among pregnant women attending antenatal care of 3.8% and 3.5%, respectively.[25 26] These prevalences are also comparable to our study. By contrast, low endemicity areas, such as the USA, report values around 0.38%.[27] These low endemicity areas have vigorously implemented hepatitis B prevention strategies, such as universal testing of all pregnant women for hepatitis B, universal vaccination at birth and universal childhood immunisation. These strategies are uncommon in intermediate and high endemicity areas.[28]

The 11.8% prevalence of hepatitis B in pregnant women found by Bayo *et al* in Northern Uganda in 2014[5] and the average national prevalence of ~10% in the general population[4] are both higher than the prevalence in our study. The prevalence found in other select populations such as health workers in Mulago Hospital is also higher than in our study (8.1%[29]). Several factors may account for this difference. The central region of Uganda is a low prevalence area, compared with other regions such as the north and east. The intensification of hepatitis B prevention programmes over the last few years in Uganda may also have contributed to a reduction in prevalence of hepatitis B in the general population and among pregnant women.

In our study, we found marital status, history of having had a hepatitis B positive family member and history of

**Table 3** Association between demographic, gynaecological, obstetric characteristics and hepatitis B status

| | Hepatitis B status | | | |
| --- | --- | --- | --- | --- |
| | Positive (n=10 (%)) | Negative (n=330 (%)) | Unadjusted OR (95% CI) | P value* |
| **Demographic characteristics** | | | | |
| Age (years); mean (SD) | 27.4 (6.2) | 26.9 (5.7) | | 0.775† |
| Age (years) categorised | | | | 0.582 |
| <20 | 1 (10.0) | 27 (8.2) | 1.00 | |
| ≥20 | 9 (90.0) | 303 (91.8) | 1.25 (0.15 to 10.21) | |
| Employment status | | | | 0.753 |
| Yes | 6 (60) | 172 (52.2) | 1.00 | |
| No | 4 (40) | 158 (47.8) | 0.73 (0.20 to 2.62) | |
| Marital status | | | | 0.004 |
| Staying with a partner | 6 (60.0) | 298 (90.3) | 1.00 | 0.014 |
| No stable partner | 4 (40.0) | 32 (9.7) | 6.21 (1.66 to 23.16) | |
| **Gynaecological/obstetric factors** | | | | |
| Weeks of amenorrhea categorised | | | | <0.001 |
| >20 | 1 (10.0) | 210 (63.6) | 1.00 | |
| ≤20 | 6 (60.0) | 48 (14.6) | 26.25 (3.09 to 223.13) | |
| Unknown | 3 (30.0) | 72 (21.8) | 8.75 (0.90 to 85.46) | |
| Mother hepatitis B positive? | | | | 0.036 |
| Yes | 1 (10.0) | 3 (0.9) | 1.00 | |
| No | 1 (10.0) | 125 (37.9) | 0.02 (0.00 to 0.48) | |
| I don't know | 8 (80.0) | 202 (61.2) | 0.12 (0.01 to 1.27) | |

*Fisher's exact p values.
†t-Test p values.

having had a blood or body fluid splash from a hepatitis B positive patient as factors independently associated with hepatitis B infection (table 5). Other factors identified as significant in previous studies, such as age,[5 27 30 31] level of education,[31 32] socioeconomic status,[33] number of sexual partners,[26 34] piercing of ears,[34] history of abortions,[25 26] history of blood transfusion[30] and history of jaundice,[35] were not significantly associated with hepatitis B infection in this study. HIV serostatus, a factor that several studies have found associated with hepatitis B infection,[30 33] did not have statistical significance in our study (table 4). This could be because the HIV prevalence in our study was generally low (table 2).

Women who did not have a stable partner were 11 times as likely to be hepatitis B positive as their counterparts who lived with partners (adjusted OR (aOR)=11.37, 95% CI 2.37 to 54.60, p=0.002, table 5). Similar results have been reported from Juba, South Sudan, although the increased risk was specific to loss of a partner (OR 4.4, 95% CI 1.4 to 13.9).[35] Chernet *et al* and Umare *et al* also found an association between multiple sexual partners and hepatitis B positivity among pregnant women in Ethiopia (aOR=6.923 and aOR=16.8, respectively).[26 34] By contrast, studies from Kenya and Nigeria did not find any significant association between marital status and hepatitis B status.[25 36] In our study, the women with no stable

partner were widowed, separated or unmarried. This increases the risk of having contracted hepatitis B from the lost partner or via risk-taking sexual behaviour.

Study participants who had a hepatitis B positive family member were almost 50 times more likely to be positive for hepatitis B than those who did not (aOR=49.52, 95% CI 6.24 to 392.85, p<0.001, table 5). Data from the CDC support this finding, showing that 3%–20% of close contacts of people with chronic hepatitis B are also infected. Screening and vaccination of all close contacts of known hepatitis B cases is therefore recommended.[37] A study from South Sudan also showed an increased risk of hepatitis B infection among pregnant women who reported a history of loss of a partner (OR=4.4).[35] This important finding underscores the risk of horizontal transmission of hepatitis B infection either sexually between partners or through contact with blood and body fluids among people staying in the same home. This can be through sharing household items such as toothbrushes, razors, syringes and needles. The risk posed by having to take care of a hepatitis B positive patient is even higher, with exposure to blood and body fluids infected with the virus. It is essential that those who live with, or take care of, a hepatitis B positive individual be fully vaccinated to reduce the risk. All hepatitis B positive mothers should be identified during antenatal care

**Table 4** Association of immunosuppression characteristics/occupational exposure risk/surgical procedures/sharp instruments and familial exposure with hepatitis B status

| | Hepatitis B status | | | |
| | Positive (n=10 (%)) | Negative (n=330 (%)) | Unadjusted OR (95% CI) | P value |
|---|---|---|---|---|
| **Immunosuppression characteristics** | | | | |
| HIV sero-status | | | | 0.261 |
| Positive | 1 (10.0) | 9 (2.7) | 3.96 (0.45 to 34.69) | |
| Negative | 9 (90.0) | 321 (97.3) | 1.00 | |
| **Occupational exposure risk** | | | | |
| Blood or body fluid splash from a hepatitis B positive patient | | | | 0.058 |
| Yes | 1 (10) | 1 (0.3) | 36.56 (2.11 to 631.85) | |
| No | 9 (90) | 329 (99.7) | 1.00 | |
| **Exposure through surgical procedures/sharp instruments** | | | | |
| Ever had a blood transfusion | | | | 0.424 |
| Yes | 1 (10.0) | 17 (5.2) | 2.05 (0.24 to 17.09) | |
| No | 9 (90.0) | 313 (94.8) | 1.00 | |
| Ever pierced your ears | | | | 0.411 |
| Yes | 7 (70.0) | 268 (81.2) | 1.00 | |
| No | 3 (30.0) | 62 (18.8) | 1.85 (0.47 to 7.37) | |
| **Familial exposure** | | | | |
| Have a hepatitis B positive family member at home? | | | | 0.011 |
| Yes | 2 (20.0) | 5 (1.5) | 1.00 | |
| No | 6 (60.0) | 285 (86.4) | 19.00 (3.05 to 118.24) | |
| Don't know | 2 (20.0) | 40 (12.1) | 2.38 (0.46 to 12.17) | |

and vertical transmission prevented through hepatitis B vaccination of the infant within the first 24 hours after delivery.

Those who had ever had a blood or body fluid splash to mucous membranes from a hepatitis B positive patient were almost 62 times more likely to be hepatitis B positive than those who were unexposed (aOR=61.69, 95% CI 2.24 to 1701.54, p=0.015, table 5). This is a significant potential sources of hepatitis B infection either from infected patient to a healthcare worker or vice versa. A study by Ziraba *et al* has previously shown that only small percentage of the health workers in our study setting were vaccinated.[29] The WHO estimates the risk of infection with hepatitis B after a single percutaneous exposure to be 6%–60%, which is higher than hepatitis C (2%) or HIV (0.1%–0.3%).[38] It is extremely important to encourage vaccination of healthcare workers and people in contact with hepatitis B positive patients.[39]

A potential limitation of our study is that we performed it in a national referral and teaching hospital, meaning it may not accurately reflect the situation in other settings. The limited number of hepatitis B positive patients in our study also leads to increased uncertainty regarding the magnitude of risk conferred by each factor studied.

## CONCLUSION

The prevalence of hepatitis B infection among pregnant women attending antenatal care clinic in Mulago Hospital is of intermediate endemicity. We found that several independent risk factors were associated with increased likelihood of being hepatitis B positive: marital status, history of having had a hepatitis B positive family member and history of a blood or body fluid splash to mucous membranes from a hepatitis B positive individual. Factors such as age, HIV status, history of blood transfusion, piercing of ears and social status were not associated with the likelihood of being hepatitis B positive.

## RECOMMENDATIONS

Because hepatitis B is of intermediate endemicity in our population, we recommend testing all pregnant women for hepatitis B during antenatal care. All babies born to hepatitis B positive mothers should undergo the full schedule of hepatitis B vaccination, with or without hepatitis B immunoglobulin starting at birth.

Since most of the adult population of Uganda were not vaccinated in childhood, we recommend screening and adult vaccination for high-risk groups, such as healthcare workers and people living or taking care of hepatitis B

**Table 5** Multivariable logistic regression model showing factors associated with hepatitis B virus infection among study participants

| Characteristics | Adjusted OR (95% CI) | P value |
|---|---|---|
| Age (years) categorised | | |
| <20 | 1.00 | |
| ≥20 | 0.27 (0.01 to 5.31) | 0.387 |
| Marital status | | |
| Staying with a partner | 1.00 | |
| No stable partner | 11.37 (2.37 to 54.60) | 0.002 |
| Have a hepatitis B positive family member at home | | |
| No | 1.00 | |
| Yes | 49.52 (6.24 to 392.85) | <0.001 |
| Don't know | 4.25 (0.71 to 25.35) | 0.113 |
| Blood or body fluid splash from hepatitis B positive patient | | |
| No | 1.00 | |
| Yes | 61.69 (2.24 to 1701.54) | 0.015 |

positive individuals to reduce the likelihood of horizontal transmission.

We further recommend vaccination of pregnant women against hepatitis B, which is not contraindicated in pregnancy. There should be clear referral pathways for pregnant women who are hepatitis B positive to access specialist care for further serological, virologic and hepatological assessment and management, including long-term monitoring.

Finally, we recommend a larger multicentre study in Uganda that includes tests for viral load and anti-hepatitis B core antibody. This will yield more important information regarding important exposures, risk of vertical transmission and more generalisable risk factors.

**Acknowledgements** We thank all senior members of the Department of Obstetrics and Gynaecology, Makerere University and members of the Directorate of Obstetrics and Gynaecology, Mulago Hospital for the support and direction during the study. We thank Associate Professor Annette Nakimuli for her tremendous input to this study and providing a conducive environment for us to carry it out. Great thanks to Mr Godwin Anguzu, who offered technical input during data analysis. Finally, great thanks to the whole dedicated data collection team in the Mulago Hospital antenatal care clinic, Kawempe site.

**Collaborators** Erasmus Okello Erebu Sam Ononge Annette Nakimuli.

**Contributors** SPK developed the study's concept, designed the study methods and also took part in data collection, analysis and interpretation. PN revised the manuscript critically and made amendments for important intellectual content. JKB contributed significantly to the intellectual content of the manuscript and also gave the final approval of the version submitted. All the authors have agreed to be accountable for all aspects of the work in this study.

**Funding** DELTAS Africa Initiative grant #DEL-15-011 to THRiVE-2 supported this work. The DELTAS Africa Initiative is an independent funding scheme of the African Academy of Sciences (AAS)'s Alliance for Accelerating Excellence in Science in Africa (AESA) and supported by the New Partnership for Africa's Development Planning and Coordinating Agency (NEPAD Agency) with funding from the Wellcome Trust grant #107742/Z/15/Z and the UK government. The views expressed in this publication are those of the authors and not necessarily those of AAS, NEPAD Agency, Wellcome Trust or the UK government.

**Competing interests** None declared.

**Patient consent for publication** Not required.

**Ethics approval** Ethical approval was granted by the Department of Obstetrics and Gynaecology, Makerere University, and Makerere University School of Medicine Research and Ethics Committee (#REC REF 2018–119), Mulago Hospital Research and Ethics Committee (#MHREC1519) and the Uganda National Council for Science and Technology (#HS257ES), before any collection of samples or data.

**Provenance and peer review** Not commissioned; externally peer reviewed.

**Data availability statement** All data relevant to the study are included in the article or uploaded as supplementary information. Raw data are accessible on: https://drive.google.com/drive/folders/1wc0C3bP4Qge0OKtG4z-zXC6pbD1F3nsL?usp=sharing.

**ORCID iD**
Simon Peter Kayondo http://orcid.org/0000-0003-2087-1295

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
