## [Reviewer comments · BMJ Open]

ARTICLE DETAILS

TITLE (PROVISIONAL)	Prevalence of Hepatitis B Virus infection and associated risk factors among pregnant women attending antenatal clinic in Mulago Hospital, Uganda: A cross-sectional study.
AUTHORS	Kayondo, Simon Peter; Byamugisha, Josaphat; Ntuyo, Peter

VERSION 1 - REVIEW

REVIEWER	Baba maiyaki Musa Bayero University Kano Nigeria
REVIEW RETURNED	11-Feb-2020

GENERAL COMMENTS	My understanding is the original work might have been that of the master thesis of one of the authors. It appears some more work is needed to convert the narrative to what would suit publication in a journal of BMJ repute. The language of the manuscript needs significant improvement, and typos should be addressed. I am a bit concerned with the wide confidence interval around the prevalence value. It may suggest perhaps a discrepancy between stated sample size and actual sample size. I am also surprised that prevalence among HIV positive persons is quite low (only one person). I would also want to draw your attention that the hospital is a referral one and indeed offering services to migrants. It is likely that prevalence from such a hospital may not reflect the true prevalence of that vicinity. Accordingly, external validity might be a concern here.
--

REVIEWER	Ousman Nyan School of Medicine & allied Health Sciences University of The Gambia PO Box 1616 Edward Francis Small Teaching Hospital Independence Drive Banjul The Gambia
REVIEW RETURNED	19-Feb-2020

GENERAL COMMENTS	1, Title: This is unduly long and the initial part can be deleted/modified as the objectives make clear that this study is focused on determining prevalence and associated risk factors and
--

	does little to address further serological, virological and hepatological assessment of screen-positive individuals. 2. Abstract: is adequately written 3. Introduction: References should be clearly provided for the HBV prevalence of 6.1% in WHOAFRO and HBV infection and its associated sequelae in sub-Saharan Africa. The history and status of introduction of HBV vaccine in Uganda should be helpful background information. 4. Methods. Regarding the sample size estimation, explanation should be provided for the rationale for using the prevalence data from N Uganda rather than that previously reported from Mulago Hospital itself. The assumption of stratifying by age=$<20y$ and $>20y$ is not supported by the gross imbalance of recruited participants as indicated in the results (Table 1). The under 20y are grossly under-represented, potentially putting strain on the statistical analysis. Generalizability is limited as only speakers of English and Luganda were recruited, in spite of the claims of targeting a nationally-representative sample from the country's National Hospital. The Laboratory test had quality control checks-but nothing is reported on the actual performance of the tests, barring what is reported (without referencing) by the (manufacturer's?) brochure. Appropriate ethical clearance and standards of care (with appropriate counselling, referral and provision of vaccine to the newborns to the HepBsAg+ve participants. Results:These are adequately presented in Tables 1-5. Discussion: The 2016 study reported from Mulago Hospital (before transfer to the new - reportedly less privileged site) had -50% unemployed participants, with -27% under 20y and a wider socio-demographic profile: the discussion on this should be the starting point on what additional contribution the present study is providing at the 'home' level. The fact that the vast majority of the HepBsAg+ve pregnant women present to clinic early (<20 weeks of amenorrhoea) should also attract some reflection. Finally, the recommendations on provision active vaccination and prophylaxis are spot on, but they do not directly speak to the need for further serological, virological and hepatological assessment and management, including long term monitoring, of screen-positive individuals.
--	--

VERSION 1 – AUTHOR RESPONSE

Reviewer(s)' Comments to Author:

Reviewer: 1

Reviewer Name

Baba maiyaki Musa

Institution and Country

Bayero University Kano

Nigeria

Please state any competing interests or state 'None declared':

None

Please leave your comments for the authors below

My understanding is the original work might have been that of the master thesis of one of the authors. It appears some more work is needed to convert the narrative to what would suit publication in a journal of BMJ repute. We have made some revisions to the style of language to address this. We enlisted the help of a scientific writing editor too. (Whole document).

The language of the manuscript needs significant improvement, and typos should be addressed. We involved a professional medical writer to help address this. (Whole document).

I am a bit concerned with the wide confidence interval around the prevalence value. It may suggest perhaps a discrepancy between stated sample size and actual sample size. The confidence intervals in our study were generally wide, perhaps due to the few positive counts. However, we strongly believe that the confidence interval around the prevalence value is appropriate, and reflective of the true situation, as determined by our study design. And, the calculated sample size, and the final sample size was the same.

I am also surprised that prevalence among HIV positive persons is quite low (only one person). This was an interesting finding in our study. We believe this could be because of the generally low prevalence of HIV in our study. ((Included in the discussion, pages 14 & 14)

I would also want to draw your attention that the hospital is a referral one and indeed offering services to migrants. It is likely that prevalence from such a hospital may not reflect the true prevalence of that vicinity. Accordingly, external validity might be a concern here. We do agree with this observation, and have captured it as a limitation. (Pages ii & 15)

Reviewer: 2

Reviewer Name

Ousman Nyan

Institution and Country

School of Medicine & allied Health Sciences

University of The Gambia

PO Box 1616

Edward Francis Small Teaching Hospital

Independence Drive

Banjul

The Gambia

Please state any competing interests or state 'None declared':

None declared

Please leave your comments for the authors below

1, Title: This is unduly long and the initial part can be deleted/modified as the objectives make clear that this study is focused on determining prevalence and associated risk factors and does little to address further serological, virological and hepatological assessment of screen-positive individuals. The title has been revised accordingly. (Title page).

2. Abstract: is adequately written

3. Introduction: References should be clearly provided for the HBV prevalence of 6.1% in WHOAFRO and HBV infection and its associated sequelae in sub-Saharan Africa. This is captured in the first paragraph of the introduction section. (Page 1). The history and status of introduction of HBV vaccine in Uganda should be helpful background information. This has been attended to in the revised manuscript. (Page 2)

4. Methods. Regarding the sample size estimation, explanation should be provided for the rationale for using the prevalence data from N Uganda rather than that previously reported from Mulago Hospital itself. The prevalence reported from the study done in Mulago hospital (the old site), of 0.9%

was unusually lower than most, determined from all other all previous studies, including the survey for national estimates. In addition, this study excluded a wide range of participants: - those below 14 years, and those above 43 years, as well as the bed-ridden and those with obstetric emergencies. And due to the paucity of data around the country on prevalence of Hepatitis B virus infection among pregnant women, the next closely related study was the one done in Northern Uganda. The assumption of stratifying by age= $\leq 20y$ and $>20y$ is not supported by the gross imbalance of recruited participants as indicated in the results (Table 1). The under 20y are grossly under-represented, potentially putting strain on the statistical analysis. In the Northern Uganda study, used in the calculation of sample size, maternal age less than twenty years was found to be independently associated with likelihood of having Hepatitis B virus infection. We therefore adopted the same stratification, to test its association. Interestingly, under 20-year olds were under represented, possibly because of a low prevalence in this group, in this study setting. Generalizability is limited as only speakers of English and Luganda were recruited, in spite of the claims of targeting a nationally-representative sample from the country's National Hospital. Luganda is the local language, and most other non-Luganda speakers could express themselves in English. The exclusion was made to make it convenient for the research assistants to deal with the interviews, and they were comfortable with English and Luganda too.

The Laboratory test had quality control checks-but nothing is reported on the actual performance of the tests, barring what is reported (without referencing) by the (manufacturer's?) brochure. The manufacture's brochure showing the performance of the test is attached as an additional document.

Appropriate ethical clearance and standards of care (with appropriate counselling, referral and provision of vaccine to the newborns to the HepBsAg+ve participants.)

Results:These are adequately presented in Tables 1-5.

Discussion: The 2016 study reported from Mulago Hospital (before transfer to the new - reportedly less privileged site) had -50% un-employed participants, with -27% under 20y and a wider socio-demographic profile: the discussion on this should be the starting point on what additional contribution the present study is providing at the 'home' level. Different findings socio-demographically. The fact that the vast majority of the HepBsAg+ve pregnant women present to clinic early (<20 weeks of amenorrhoea) should also attract some reflection. This could be as a result of the intensified sensitisation for women to attend antenatal care early, but perhaps the fact that more women are able to be diagnosed early on in pregnancy could be an opportunity for intervention. These reflections have been included in the revised manuscript. (Pages 12 & 13)

Finally, the recommendations on provision active vaccination and prophylaxis are spot on, but they do not directly speak to the need for further serological, virological and hepatological assessment and management, including long term monitoring, of screen-positive individuals. This is taken into consideration in the revised manuscript, and incorporated as a recommendation.(Page 15)

VERSION 2 – REVIEW

REVIEWER	Baba Musa Bayero University Kano Nigeria
REVIEW RETURNED	01-Apr-2020

GENERAL COMMENTS	I recommend the manuscript for publication
--

REVIEWER	Prof Ousman Nyan School of Medicine & Allied Health Sciences Edward Francis Small Teaching Hospital Independence Drive Banjul The Gambia
REVIEW RETURNED	15-Apr-2020

GENERAL COMMENTS	All previous queries/concerns have been addressed, except for the elusive issue of quality assurance evaluation of the commercial test. The manufacturer's brochure reads well but the authors had claimed to do more to assure test performance.
---

VERSION 2 – AUTHOR RESPONSE

Reviewer(s)' Comments to Author:

Reviewer: 1
Reviewer Name
Baba Musa
Institution and Country
Bayero University Kano
Nigeria

Please state any competing interests or state 'None declared':
None declared

Please leave your comments for the authors below
I recommend the manuscript for publication

Reviewer: 2
Reviewer Name
Prof Ousman Nyan
Institution and Country
School of Medicine & Allied Health Sciences
Edward Francis Small Teaching Hospital
Independence Drive
Banjul
The Gambia

Please state any competing interests or state 'None declared':
none

Please leave your comments for the authors below
All previous queries/concerns have been addressed, except for the elusive issue of quality assurance evaluation of the commercial test. The manufacturer's brochure reads well but the authors had claimed to do more to assure test performance. On page 5 in the second revision of the manuscript, two areas are highlighted indicating how we did quality control and assurance of the test. Before doing any participant sample testing, the test kit was quality controlled using known positive and negative samples, at an accredited laboratory. We also did confirmatory tests on all our positive samples at this same laboratory, as well as quality assurance tests on 5% of all our negative samples. There were no contradictory results.